# Seed Germination in *Cistus ladanifer*: Heat Shock, Physical Dormancy, Soil Temperatures and Significance to Natural Regeneration

**DOI:** 10.3390/plants8030063

**Published:** 2019-03-12

**Authors:** Luís Silva Dias, Isabel Pires Pereira, Alexandra Soveral Dias

**Affiliations:** 1Department of Biology, University of Évora, Ap. 94, 7000-554 Évora, Portugal; ipp@uevora.pt (I.P.P.); alxandra@uevora.pt (A.S.D.); 2Institute of Mediterranean Agricultural and Environmental Sciences, University of Évora, Ap. 94, 7000-554 Évora, Portugal

**Keywords:** *Cistus ladanifer*, fire effects, heat shock, imbibition, physical dormancy, seed coat, seed germination, soil temperatures

## Abstract

Seeds of *Cistus ladanifer* experience bursts of germination following fires. The effects of heat shock from 10 °C to 150 °C on seed germination were investigated by final germination plus the number of days required for germination to start and finish, and symmetry of cumulative germination. The occurrence of physical dormancy in *C. ladanifer* seeds was investigated by a variety of methods, including imbibition, scanning electron microscopy (SEM) and light microscopy, and use of dyes. The significance of responses of *C. ladanifer* seeds to fires was investigated essentially by abstracting existing literature and by using fire effects models and simulations. Parameters of germination were variously affected by heat treatments—positively in the range 80–100 °C, negatively above 130 °C. Non-dormancy was consistently found in about 30% of seeds but no evidence was obtained to support the existence of physical dormancy in the dormant fraction of *C. ladanifer* seeds. Two complementary processes seem to be in place in seeds response to fire. A direct fire-driven increase in germination of virtually all seeds in response to the appropriate heat load produced by fire or, in the absence of such heat loads, the germination of the non-dormant fraction provided that above-ground vegetation burns.

## 1. Introduction

Mediterranean-type ecosystems are fire-prone habitats where a large number of species have seed germination stimulated directly or indirectly by fires, but either way the general result is the induction of massive germination of largely dormant or quiescent soil seedbanks, especially of annuals and shrubby perennials [1]. *Cistus ladanifer* (Cistaceae) and other *Cistus* are some of those shrubby perennials that show dramatic increases of germination following fires [2,3,4].

Portugal and Spain are the main areas of distribution of *C. ladanifer*, which is also cultivated and highly appreciated as ornamental elsewhere [5,6]. This species occupies large areas, frequently as pure stands which are viewed as advanced stages of sclerophyllous forests degradation [7]. It is a pioneer species that slows or almost stops successions. Its spreading is strongly enhanced by field abandon from cultivation, overgrazing and fire [8,9], and reports of its possible classification as invasive species in South Africa appeared recently [10].

Along with other *Cistus* it is an obligate seeder [2,11,12,13] with small seeds. Seeds of *C. ladanifer* are held inside capsules and usually are very numerous, globose-polyhedric [14]. Their seed coats are made of an outer waxy layer, followed by palisade cells with strongly lignified cell walls. Between the palisade cells and the endosperm one or more layers of slightly differentiated cells occur. The endosperm is well developed, rich in starch and completely surrounds the embryo [15].

In the 1950s and early 1960s the usefulness of several *Cistus* to control erosion and replace native species less resistant to fires or faster burners was investigated in Southern California [11,16,17,18], which might explain the occurrence of *Cistus* in a number of Californian counties nowadays [19]. *C. ladanifer* was found to be the most flammable *Cistus* investigated [11]. However, in general *Cistus* appear to be much less flammable [20] than has been stated before [21].

Nevertheless, laboratory and field studies have shown that heat alone can induce very high percentages of germination in *C. ladanifer* [11,12,22,23,24,25,26]. In general only a relative small range of temperatures have been tested and attention was only paid to final germination. Thus it is unknown whether and how heat shock affects other important aspects of germination like the time needed to its start or the time needed afterwards to its completion.

The most frequent explanation for the direct effects of fire on germination of *C. ladanifer* and other *Cistus* is that heat shock causes the rupture of seed coats allowing water to reach embryos [4,22,25,27,28]. This explanation is based upon the assumption that seeds of *C. ladanifer* and other *Cistus* possess physical dormancy, their seed coats being impermeable to water [29,30]. However, a screen of literature fails to show experimental support to the assumption that *C. ladanifer* seed coat is a physical barrier to water uptake by seeds.

This study aimed to investigate three main questions. (1) What are the effects of heat shock on germination of *C. ladanifer* seeds evaluated across a wide range of heat shock temperatures not only by final germination but also by time required for germination to start and to finish, and by the shape of cumulative germination over time? (2) Is there evidence that the response of *C. ladanifer* seed germination to heat shock results from increased water uptake due to mechanical fracture of seed coat? (3) Given soil temperatures during fires what might be the ecological significance of seed responses to heat shock to the renewal of *C. ladanifer* stands?

## 2. Results

### 2.1. Experiment 1: Effects of Temperature and Light Regime on Final Seed Germination of *Cistus ladanifer*

Data of final germination of the two sets of seeds incubated in two different occasions under light and constant 20 °C were compared, found not significantly different (*t*_6_ = 0.12, *P* = 0.912), and pooled. Final germination ranged from 21.6 ± 5.0% to 37.9 ± 7.6%, at constant 30 °C and 15 °C respectively, both with photoperiod (Appendix A). Homoscedasticity among samples was present and no significant differences were found among the final germination of treatments with photoperiod, regardless of seeds being incubated under constant or alternate temperatures (*F*_7,28_ = 1.58, *P* = 0.182), neither between constant and alternate temperatures (*t*_34_ = 1.12, *P* = 0.269), nor between light and dark treatments (*t*_10_ = 0.28, *P* = 0.788).

### 2.2. Experiment 2: Effects of Heat Shock on Seed Germination of *Cistus ladanifer*

Parameters fitted for the relationship between heat shock lasting 15 min and final germination using Equation (4) were *x* = –48.031, *z* = 0.395 for the segment 10–70 °C; *x* = –22.596, *z* = 2.637 for the segment 70–90 °C; *x* = 24.998, *z* = 2.592 for the segment 90–140 °C but data for the 130 °C treatment had to be omitted from the regression analysis. *R*^2^-values were 0.777, ≈1, and 0.985 respectively. All predicted mean values of final germination lied within the 95% confidence intervals of observed means. Combining the tree models (overall *R*^2^ = 0.960) final germination increased slowly with heat treatments (Figure 1a and Appendix A) from the 10 °C treatment (31.7 ± 5.0%) to the 70 °C treatment (48.4 ± 6.9%). From then on the response was more intense, with all seeds germinating in the 90 °C treatment. For higher temperatures, final germination decreased until the 120 °C treatment (26.3 ± 1.4%). No germination was recorded in 140 °C and 150 °C treatments. Seeds of these two treatments and nongerminated seeds treated with 120 °C and 130 °C were preyed by microbes and rotted.

The three-term Weibull equation (Equation (3)) could be fitted to cumulative germination over time in 53 replicates, including at least two in every heat treatment (Appendix A). *R*^2^ ranged between 0.167 and ≈1 (mean 0.828 ± 0.027). Regressing parameters of Weibull equation, including derived ones, to heat treatments was only achieved in lag of germination.

Lag of germination *l* ranged from 2.0 ± 0.03 days at 90 °C to 9.2 ± 0.9 days at 120 °C (Figure 1b and Appendix A). The power law (Equation (2)) fitted to the descending segment of *l* was highly significant (*F*_1,36_ = 10.84, *P* = 0.002, *R*^2^ = 0.583, lack of fit with *F*_7,29_ = 1.14, *P* = 0.366). Fitted parameters were *a* = 11.960 and *b* = 0.349. All predicted mean values of germination lied within the 95% confidence intervals of observed means except for heat treatments 10 °C and 50 °C.

Parallel to the dose-response approach used with final germination and lag of germination, point estimates of all parameters for 10 °C to 130 °C treatments were compared. Results for homocedasticity tests, maximum likelihood estimates of Box–Cox transformations, and simultaneous comparisons of means are summarized in Appendix B.

Significant differences were found among final germination of seeds treated with 80 °C, 90 °C, 100 °C and 130 °C and all others which did not differ among them (pooled mean of the former 38.7 ± 2.6%). Also, no significant differences were found (*t*_6_ = 0.34, *P* = 0.742) between final germination of seeds treated with 80 °C and 100 °C (pooled mean 87.4 ± 4.4%).

Significant differences were found among lag of germination of seeds treated with 40 °C, 70 °C, 80 °C and 90 °C, which did not differ among them (pooled mean 2.6 ± 0.3 days) and all others which did not differ among them also (pooled mean 5.5 ± 0.4 days).

The number of days necessary to attain final germination minus the number of days required to its beginning (*D*_100_) ranged from 8.1 ± 0.7 days at 30 °C, closely followed by 90 °C (9.6 ± 0.2 days) to 21.0 ± 5.3 days at 80 °C (Figure 1c and Appendix A) with heat treatment 90 °C breaking the trend for an oscillatory behavior of *D*_100_. Significant differences were found among *D*_100_ of seeds treated with 30 °C and all others which did not differ among them (pooled mean 14.8 ± 0.9 days).

Shape of germination *c* ranged from 2.2 ± 0.4 at 100 °C to 8.4 ± 1.7 at 90 °C (Figure 1d and Appendix A). Significant differences were found among *c* of seeds treated with 90 °C and all others which did not differ among them (pooled mean 3.7 ± 0.3).

Pairwise variation of final germination, lag of germination, and duration of germination is plotted in Figure 2 and shows an inverse variation of final germination and lag of germination (Figure 2a), a tendency for a direct variation of final germination and duration of germination (Figure 2b), which essentially results in no common variation, either inverse or direct between lag and duration of germination (Figure 2c).

### 2.3. Experiment 3: Effects of Heat Shock on Seed Coat Morphology of *Cistus ladanifer*

Micrographs of representative sites of untreated seeds, of seeds treated with heat or with hexane are presented in Figure 3.

Scanning electron micrographs of untreated seeds of *C. ladanifer* externally show a continuous layer with a reticulate pattern made of irregular ridges (Figure 3a) identical to previous reports of seeds of this species [15,31]. Extraction of epicuticular waxes by hexane revealed that the surface of the internal layer is flatter with a reticulate pattern of irregular openings coincident with the ridge pattern of the outermost layer (Figure 3b). Seeds treated with heat (90 °C) resemble very closely untreated seeds except that the reticulate was smoothed and ridges were less prominent (Figure 3c).

### 2.4. Experiment 4: Weight Increase of Nongerminated Imbibing Seeds of *Cistus ladanifer*

Final germination was 32.3 ± 2.7% and was not significantly different from the mean of final germination recorded in the same experimental conditions in Experiment 1 (*t*_10_ = 1.23, *P* = 0.248) and in Experiment 2 (*t*_6_ = 0.11, *P* = 0.210), 27.3 ± 2.5% and 32.6 ± 1.2% respectively (Appendix A).

Seed weight before imbibition and incubation ranged from 0.184 mg to 0.347 mg (mean 0.282 ± 0.003 mg). With few exceptions the weight of all seeds that did not germinate and therefore could be weighed again, increased during the experiment (Appendix A), increases having been found in 83.3 ± 4.8% of nongerminated seeds.

There was a considerable overlap between the initial weights of seeds that later germinated (0.184 mg to 0.347 mg, mean 0.293 ± 0.007 mg) and those that did not (0.220 mg to 0.341 mg, mean 0.277 ± 0.004 mg), and no significant differences were found between them (*t*_84_ = 2.20, *P* = 0.050). Therefore germination seems to be independent of the size of seeds contrary to what is frequently the case in other species [29] and cannot be used as a predictor of final germination of *C. ladanifer* seeds.

Weight increases of nongerminated seeds weighed by initial seed weight before imbibition were relatively low ranging from 0.3% to 13.0%, with a mean of 4.7 ± 0.4%.

### 2.5. Experiment 5: Dye Uptake by Imbibing Seeds of *Cistus ladanifer*

Methyl violet was found inside almost all seeds (92.9 ± 4.1%). In the majority of seeds the whole internal space was dyed but not embryos (Figure 4c). Simultaneously no seed size increases were detectable in 61.6 ± 1.9% of seeds. However size visibly increased in a number of seeds in which embryos were dyed (Figure 4d).

Swollen and dyed seed embryos included, amounted to 36.3 ± 2.7% of seeds a value that was not significantly different from mean final germination of seeds treated with 30 °C in Experiment 2 (*t*_6_ = 1.21, *P* = 0.252) and Experiment 4 (*t*_6_ = 1.06, *P* = 0.330) and virtually not significantly different from final germination of seeds treated with 30 °C in Experiment 1 (*t*_10_ = 2.32, *P* = 0.050), 32.6 ± 1.2%, 32.3 ± 2.7%, and 27.3 ± 2.5% respectively (Appendix A).

### 2.6. Experiment 6: Effects of Heat Shock on Volume Increase of Imbibing Seeds of *Cistus ladanifer*

Almost all swollen seeds visibly took water in the first 72 h of incubation. In the 6th day of imbibition significant differences (*t*_6_ = 24.29, *P* ≈ 0) were found between the cumulative percentage of swollen seeds in 30 °C and 90 °C treatments, which was 31.0 ± 2.5% and 98.0 ± 1.1% respectively. Conversely, no significant differences in the percentage of swollen seeds treated with 30 °C were found between this experiment and Experiment 5 (*t*_6_ = 1.64, *P* = 0.152).

Final germination in the same heat treatments of 30 °C and 90 °C in Experiment 2 was 32.6 ± 1.2% and 100 ± 0% (Appendix A). No significant differences were found neither in 30 °C heat treatment (*t*_6_ = 0.58, *P* = 0.583) nor in 90 °C heat treatment (*t*_6_ = 1.73, *P* = 0.182) between the percentages of swollen seeds in this experiment and the final germination in Experiment 2.

### 2.7. Depth of Emergence of *Cistus ladanifer* Seedlings and Soil Temperatures

#### 2.7.1. Depth of Emergence of *Cistus ladanifer* Seedlings

Maximum hypocotyl length in *C. ladanifer* seedlings growing in filter paper was reported to be 22.3 mm [26]. However, from Equation (1) and from data abstracted from literature, maximum depth for successful germination of *C. ladanifer* in soil would be lesser, only 18.9 mm [26]. Using Equation (1) and data from individual seed weight of Experiment 4, maximum depth for successful germination of *C. ladanifer* in soil would range from 11.2 mm to 19.2 mm (mean 17.7 ± 0.1 mm, median 17.8 mm). Heavier seeds have been reported thus increasing the depth from which successful germination might occur to 39.6 mm [15] or to a suspiciously deep 56.0 mm [32], unless mass data provided by the latter authors refer to a sample of 25 seeds and not to a single seed in which case maximum depth would reduce to a more reasonable 19.1 mm.

#### 2.7.2. Soil Temperatures during Fires

Peaks of temperature of 250 °C can be found during about four minutes at soil surface in a typical chaparral during low burn and around 80 °C during seven minutes at 2.5 cm depth. Higher peaks were found in moderate and intense burn reaching 430 °C and 680 °C during about two minutes at soil surface, 170 °C and 190 °C during three and five minutes at 2.5 cm depth respectively [33].

During fire in late spring in a *Quercus coccifera* garrigue free from fire in the previous 20 years, temperatures higher than 200 °C with a maximum of 250 °C lasted more than two minutes at soil surface while in early autumn temperatures quickly reached 400 °C and stayed above 200 °C for more than three minutes. At 2.5 cm depth a maximum of 125 °C was registered in late spring half minute after the passage of fire, then dropped to 100 °C after another half minute, stayed unchanged during the following 5 min, then descended quickly to 80 °C followed by a slower descent to around 60 °C, while in early autumn temperature exceeded 100 °C for almost five minutes with a peak at 150 °C, then steadily declined [34].

Temperatures at soil surface during fires may reach 200 °C, while temperatures in the soil profile strongly depend on fire intensity. In light fires temperatures are less than 55 °C at 3 cm depth but the same temperatures can be found at 10 cm depth during very intense fires [35].

Time-course curves for soil temperatures after fires are also available for 50-years-old *Adenostoma* stands burned in the end of November. At soil surface temperatures in excess of 200 °C (maximum exceeding 800 °C) last approximately nine minutes. At 2 cm depth the maximum temperature was 140 °C, temperatures above 100 °C lasting approximately two hours [36].

Similar data was provided for fire temperatures in 12-years old *Ulex parviflorus* gorse shrublands. At soil surface temperatures exceeding 200 °C were observed, with a maximum of 409 °C, during approximately 15 min. At 1 cm depth maximum temperatures ranged from 55 °C to 229 °C, temperatures above 200 °C likely to last approximately five minutes then dropping to less than 60 °C after 20 min [37]. Graphical interpolation shows that mean maximum temperature at 2 cm depth would be around 70 °C. These data were collected in autumn after the first rainfalls and higher temperature values would probably be found if soil was dry as usually happens when non-experimental fires are involved.

Nevertheless, ten years after in the same gorse shrublands, maximum temperature at 1-cm depth during an experimental fire ignited in June never exceeded 100 °C with temperatures higher than 60 °C lasting two minutes or less while in *Cistus albidus* stands lower temperatures were found at 1 cm depth despite that the latter species generally has a greater fuel load than *U. parviflorus* [38].

Profiles of soil temperatures during fires were simulated using a first-order fire effects model [39] and three types of fuel load models selected according to [40]. Fuel load models were FLM015, FLM054 and FLM066, corresponding to shrubs, nonsagebrush with low, medium and high total loads respectively. Pacific West, summer, very dry moisture conditions and loamy-skeletal soil were selected and default values for natural fuel type were always used except that duff load and duff depth were set to zero. Lethal and near-lethal temperatures would be found at soil surface for about three minutes under FLM015 simulation which has litter and shrub loads very similar to those found in mature stands of *C. ladanifer*. At or below 1-cm depth soil temperatures would never be high enough to stimulate *C. ladanifer* seed germination. Increasing litter and shrub loads, FLM054 and FLM066, lethal temperatures would be found deeper and during more time, noneffective temperatures also deeper, and stimulatory temperatures in a small interval around 2-cm depth.

## 3. Discussion

### 3.1. Responses of Seed Germination to Temperature and Light Regime

Final germination of *C. ladanifer* seeds was unaffected by light and by constant and alternate temperatures of incubation in the range 10 °C to 30 °C. Similar results have been reported for *C. ladanifer* and other *Cistus* [3,22]. However, it was also reported that incubation at 15 °C under dark might favor final germination of *C. ladanifer* seeds [22].

### 3.2. Responses of Seed Germination to Heat Shock

The results of final germination of *C. ladanifer* seeds in response to heat shock reported here show only minor and negligible differences from similar bioassays reported in literature [22,23] and fundamentally leaves the overall picture provided by them unchanged. However the wider range of temperatures tested and the dose-response approach adopted provide a more detailed understanding of such responses.

Final germination increased with temperature until 90 °C to decrease thereafter. However no significant differences were found in final germination of seeds treated with 70 °C or less, where a non-negligible fraction of around 40% of seeds germinated.

Nevertheless heat shock alone increased final germination of *C. ladanifer* with a sharp increase to almost 90% at 80 °C. From equations fitted to data of Experiment 2 final germination would exceed 95% for heat treatments between 83 °C and 97 °C. The maximum was observed at 90 °C, a treatment where all seeds germinated. The viability of *C. ladanifer* seeds used in this experiment was not investigated per se but the high germination rates observed in seeds treated with temperatures between 80 °C and 100 °C makes a strong argument against considering lack of viability as an explanation for failure of germination in the other heat treatments. 

At 140 °C or above all seeds were rotten and heavily preyed by microbes which might result from thermal destruction or inactivation of phytoncides known to be present in seeds of *C. ladanifer* [31,41].

Heat shock also affected other aspects of germination. The reduction of lag of germination, meaning a significant trend for a faster initiation of germination with increasing temperatures was found in heat treatments below 100 °C. Seeds treated with 90 °C were the fastest, initiating germination only less than two days after incubation, slightly less than in seeds treated with 80 °C, bordering the status of very fast germinating seeds [42,43]. Overall, two major groups could be identified in relation to the time necessary for germination to start. One group of fast seeds largely coincident with the group of high final germination rates the exception located at heat shock 100 °C, and one group of slow seeds essentially including heat treatments below 80 °C and above 100 °C.

Time necessary to attain final germination after the lag phase and shape of germination were almost never affected with the pooled mean for the latter excluding the 90 °C treatment neared the upper limit of symmetric distribution. Therefore factors governing the germination of *C. ladanifer* appear to act additively or quasi-additively across almost all heat treatments. By the contrary the negative asymmetric distribution in seeds treated with 90 °C implies that factors governing germination after that treatment act nonadditively [44].

In short, final germination of *C. ladanifer* is enhanced by heat shock in the range 80–100 °C with maximum germination at 90 °C, which is also the heat shock treatment responsible for the fastest initiation of germination and for one of the two fastest completions of germination.

This range of optimal temperatures for final germination is not exclusive of *C. ladanifer* and is found in a variety of other species from fire-prone ecosystems [26,45,46,47,48,49,50] strongly suggesting that it represents what may be a generalized adaptive response to recurrent fires. However the discussion of whether it is in fact an adaptation or an exaptation [51,52,53,54] is beyond the scope of this paper.

### 3.3. Responses of Water Uptake by Seeds to Heat Shock

The explanation more frequently presented in the literature to account for the promotion of seed germination of *C. ladanifer* by fire involves the rupture of seed coats by heat shock presumably because seed coats in *C. ladanifer* are impermeable to water and act as a barrier to its uptake [15,22,23,24]. Experimental research has evidenced that such process is in place in a number of other species [55,56,57,58,59] with conflicting evidences in others [60,61]. Nevertheless a screen of literature fails to show experimental support to that assumption in *C. ladanifer*. Only anecdotal and inconclusive evidence exist, derived from the structure and composition of seed coats [15,31] or from experiments where seeds were scarified with sulphuric acid [23,28], razor blades or sand paper [3,26,62,63]. Scarification was found to have smaller effects on *C. ladanifer* germination than heat shock [23,26] while in other *Cistus* smaller [28], equal [62] or larger [3] effects have been found.

Contrary to what should be expected from the hypothesis of seed coat rupture, no cracks or fractures were found in SEM observations as a result of heat shock by 90 °C during 15 min. Even the reticulate pattern of openings below the waxy external layer of seed coats, and seemingly coincident with ridges of the latter, are not made visible by heat shock.

However, the small number of seeds examined and the practical impossibility of examining the entire seed coat prevent definitive conclusions on whether or not heat shock result in mechanical rupture of seed coats, which is presumed to be one pre-requisite for water uptake of seeds of *C. ladanifer*.

It should be emphasized that the putative impermeability of seed coats in *C. ladanifer* can only affect a fraction of the seeds. In fact, as repeatedly shown in this and other studies, provided that water is available, at least around one third of seeds of this species complete germination in a relatively short period without any treatment. In addition, percentages of germination as high as 60% were reached when seeds of *C. ladanifer* were treated with water solutions of a variety of nitrogenous compounds, and as high as 80% when treated with water suspensions of charred wood [64] despite that seeds were never subjected to temperatures above 24 °C. The almost inevitable conclusion is that at least 80% of those seeds, a large part of which are usually presumed to have physical dormancy, took water despite that no rupture of their seed coats happened. As far as we know this is a point that has never been addressed before but indicates that dormancy in *C. ladanifer* seeds can hardly be related with seed coats.

Most part of nongerminated seeds (83%) increased weight after five weeks of imbibition in the absence of heat shock, but a total of 11 seeds lost weight after imbibition (Appendix A), a result that has been occasionally reported for other species, namely *Amaranthus hybridis* [65] and might result from the loss of seed materials to water [66]. Such losses might be more common than reported in literature if individual seeds as reported here, instead of batches, as usually done, were weighed. Our data support this statement in the case of *C. ladanifer*. The more frequent weight increases vastly superseded weight losses. Total weight loss of nongerminated seeds amounted to 0.065 mg which was vastly exceeded by the total weight gain of 0.682 mg.

However weight increases by imbibition noticeably larger than the relatively small increases we found with *C. ladanifer* were reported in other species without effective passage of water through seed coat [67]. However the use of methyl violet showed that with few exceptions water uptake in *C. ladanifer* is not prevented by seed coats because the dye was found inside 93% seeds. Additionaly seed imbibition was relatively fast because the whole experiment was completed in 72 hours.

Despite being found inside almost all seeds, with few exceptions methyl violet was only found inside embryos of seeds visibly swollen. In addition the percentage of swollen seeds with dyed embryos essentially coincided with the percentage of non-dormant seeds found throughout this study. Assuming that the movement of methyl violet is a reliable indicator of water movement, the absence of dye in embryos of non-swollen seeds suggests that some type of barrier to the movement of water exists in dormant seeds of *C. ladanifer* somewhere between the endosperm and the embryo, thus preventing germination. Such a barrier would be eliminated by heat shock thus increasing the imbibition of embryos.

Finally the swelling of seeds was assumed as an indicator that water had been taken by embryos, and an almost perfect agreement was found between percentages of swelling in seeds treated with 30 °C or with 90 °C and percentages of final germination of seeds treated in the same manner.

Nevertheless, and whatever the pathways for water entry our results lead to the conclusion that almost all seeds imbibe in the absence of any treatment. Thus the seed coat is not a barrier to water uptake by *C. ladanifer* seeds and the widely recorded stimulation of germination by heat shock is not the result of increased permeability to water brought by mechanical rupture of seed coat. Further research is necessary to elucidate the exact processes responsible for dormancy breaking in seeds of *C. ladanifer*.

### 3.4. Ecological Significance of Seeds Response to Heat Shock

The evaluation of the ecological significance of the response of seed germination of *C. ladanifer* to heat shock requires some knowledge on whether temperatures tested in heat shock experiments including lethal ones are found in soils during fires, and on maximum depths allowing successful emergence of *C. ladanifer* plantlets.

Nevertheless, an important fraction of *C. ladanifer* seeds have the ability to germinate in the absence of heat shock at temperatures likely to be found in soil during winter months. Seeds of *C. ladanifer* are held in capsules, broadly 500–1000 seeds per capsule [24], which can easily result in hundreds of thousands of seeds being added every year to the soil seed bank by a single plant, especially in plants growing in dense stands, very close to conspecifics [68] setting the basis for a large recruitment of new seedlings. Field data supports this conclusion even if at lower rates than could be expected from laboratory results. In fact, the number of *C. ladanifer* seedlings in cut plots can be as much as 20% of the number of seedlings in burnt plots [69]. However, when vegetation is present the contribution of this fraction to the appearance of new individuals is likely to be negligible in the field because seedling recruitment in *C. ladanifer* and other *Cistus* was found to be almost absent in unburned areas under mature canopies [12,13] probably because seedling survival is prevented by canopy shading or because seed germination was previously auto-inhibited by allelopathins [70].

The maximum depth for successful emergence of Mediterranean grassland species as been set at 10 mm [71] and seeds of Mediterranean species are frequently positively photoblastic requiring light to germinate [72]. However such requirement is apparently absent in *C. ladanifer* seeds, which might allow them to germinate from depths below 10 mm. The maximum elongation of *C. ladanifer* hypocotyls growing in filter paper under dark was found to be 22.3 mm [26], which might be an overestimation given the reduced resistance to elongation in these experimental conditions. Therefore, and according to our data and to literature, 20 mm can be assumed to be the maximum depth from which seeds of *C. ladanifer* can successfully germinate and reach soil surface. Some reports make higher depths conceivable but if proven true they would be so deep that the insulation provided by soil would prevent fire-driven increases in seed germination. Nevertheless fires may remove soil from the surface bringing deeper seeds to shallower depths, an indirect but frequent post-fire effect in the Mediterranean basin, its intensity increasing with fire intensity [73]. Thus recruitment of a large number of new plants would still be possible because about one third or more of seeds are able to germinate without heat shock.

Fire effects on soil are known to be extremely variable and heterogeneous either at soil surface or at deeper levels [36] and it has been suggested that lethal temperatures occur in soils during fires but generally confined to the soil surface [45,74]. To our knowledge no data exist on time-temperature distributions in soil during fires in *C. ladanifer* stands. However, data from communities with comparable characteristics provide useful cues to evaluate whether temperatures tested in heat shock experiments occur in the soil during fires.

A search in literature reveals a considerable variability in soil temperatures and depth distribution during fires, even if only communities similar to *C. ladanifer* stands are considered. Nevertheless a general picture emerges, in which at soil surface all or near all seeds would be killed by temperatures resulting from the passage of fire. Depending on the speed of burn and on the fuel load present lethal heat loads can occur at increasing depths [23,33] suggesting that at least at small scales, all seeds located at soil depths shallow enough to allow the renewal of *C. ladanifer* stands would be destroyed by fire and no renewal of *C. ladanifer* stands would occur unless the removal of superficial soil layers referred above would operate and allow the successful germination of seeds.

Finally it is conceivable that heat loads resulting from the passage of fire might be too low to enhance the germination of seeds of *C. ladanifer* but nevertheless burn all or nearly all herbs and shrubs aboveground. In such event, the regeneration of *C. ladanifer* stands would rest in the non-dormant fraction of its seedbank which, as noted above, is large enough to make it possible, a capacity that might lend some arguments to the debate on the adaptation or the exaptation nature of the responses of *C. ladanifer* seeds to heat shocks brought by fire.

In short, a sizable fraction of *C. ladanifer* seedbanks is likely to be destroyed by fire and only seeds buried in a very narrow range of depth would be stimulated by the appropriate heat loads during fires. Nevertheless, as remarked above fire is a phenomenon extremely variable and heterogeneous either at soil surface level or through the soil profile and in its passage a variety of heat loads are produced and seeds of *C. ladanifer* would respond in a variety of ways depending on their dormancy characteristics. Therefore, the ‘window’ of germinability [75] might be highly expanded and respond to fire characteristics and soil temperatures in a very flexible mode.

After being destroyed by fire, regeneration of *C. ladanifer* stands completely depends upon soil seedbanks. *C. ladanifer* is known to germinate very quickly after the first rains post-fire with germination occurring predominantly during the first year. As discussed above such recruitment can result from the germination of seeds whose dormancy has been removed by fire, fire being a sufficient but not necessary condition because in all likelihood the fraction of non-dormant seeds would be enough to post-fire recovery of *C. ladanifer* stands. Mortality is lower for first-year seedlings and is closely related to weather conditions like frost and dry periods occurring soon after germination took place. Nevertheless even the seedling cohorts with higher survival may have as much as 50% mortality after less than six months [2]. Anyway, in 3–4 years plants reach maturity, seed production resumes [68] and seedbanks replenish.

## 4. Materials and Methods

### 4.1. Seed Collection

Seeds were collected in early summer near Valverde (approximately 38°31’58’’ N, 8°01’33’’ W), Évora, Southern Portugal. During the 30-year period 1971-2000, yearly rainfall in the area slightly exceeded 600 mm with a very dry period (months with less than 1.5% of yearly rainfall) in July and August. Monthly mean air temperatures (maximum/minimum) were 30.2 °C/16.3 °C and 12.8 °C/5.8 °C in the hottest (July or August) and coolest (January) month respectively. The climate of Évora region is Mediterranean, mild with dry hot summers, Csa using the Köppen classification system [76].

Ripe capsules of *C. ladanifer* (subsp. *ladanifer*) that showed no evidence of predation were harvested from two, maximum three neighboring plants about 1.5 m tall located in the edge of a relatively dense stand of *Eucalyptus globulus* facing East. Seeds were extracted from capsules, stored in paper envelopes, and kept under dark at room temperature.

### 4.2. Experiments 1 and 2: Germination Experiments

Four replicates of 25 seeds of *C. ladanifer* were used for each treatment in all germination experiments. Seeds were sown in 10-cm ∅ glass Petri dishes with Whatman No. 1 filter paper, wetted with 3.5 mL of distilled water and as needed thereafter, incubated in light- and temperature-controlled incubators, checked 2-3 times a week and removed when germinated. Germination was scored when the embryo, normally the radicle, protruded from the seed coat and exceeded the largest dimension of the seed [77]. Experiments were continued until no new germination had been recorded for seven consecutive days.

#### 4.2.1. Experiment 1: Effects of Temperature and Light Regime on Final Seed Germination

This experiment was aimed to assess the responses of final germination of seeds of *C. ladanifer* to different temperatures, to constant and alternate temperatures, and to the presence or absence of photoperiod. Seeds were assayed at different temperatures regimes and light conditions to test whether germination varied across a range of incubation temperatures and light regimes. Incubators were set at constant temperatures ranging from 10 °C to 30 °C (at 5 °C intervals) and at alternating temperatures under 16/8 h cycles at 10/20 °C, 15/25 °C and 20/30 °C, and a photoperiod of 8 h simultaneous with the higher temperature in the alternating temperatures. Two additional sets of four Petri dishes each were later prepared, one wrapped in aluminum foil and black plastic, and incubated in dark under constant 20 °C (dark treatment) the other incubated under constant 20 °C and a photoperiod of 8 h (light treatment). Germination was monitored during 5 weeks and checked during the light period. In the dark treatment seeds were examined in a dark chamber under dim green-filtered light.

#### 4.2.2. Experiment 2: Effects of Heat Shock on Seed Germination

This experiment aimed to assess the effects of heat shock on germination of *C. ladanifer* seeds, described by final germination and by lag, rate and shape of germination, thereby considering the whole process and separately examining its end, represented by final germination, and the path taken to attain it [78]. Seeds were treated under dry conditions during 15 min with constant temperatures ranging from 10 °C to 150 °C (at 10 °C intervals) in refrigerated cabinet or electric oven. Heat treatments were applied for 15 min because it is unlikely that seeds of *C. ladanifer* might successfully emerge in soils below 25 mm [26], a depth at which heat waves were found to last 15 min [34] in ecosystems closely resembling those dominated by *C. ladanifer*. Typically, for the 90 °C treatment, opening the oven to introduce the seeds caused temperature to drop less than 2 °C, with the intended temperature being restored after 2–3 min. After heat treatments, seeds were left in closed and dry environment for no more than one hour, transferred to Petri dishes, wetted and incubated under constant 20 °C with a photoperiod of 8 h. Logistic constraints prevented us to perform all bioassays simultaneously and when the second series of bioassays was done some randomly selected treatments were repeated to control the consistency of responses. The 120 °C and 130 °C were also repeated because when first performed no germination was found in too many dishes to allow for fitting Equation (3) to at least two samples and again some randomly selected treatments were repeated to control the consistency of responses. Therefore two sets of four replicates each were separately tested in treatments 20 °C, 40 °C, 60 °C, 70 °C, 110 °C, 120 °C, and 130 °C and germination data pooled together (Appendix A). Germination was monitored for 6 weeks during the light period.

### 4.3. Experiment 3: Effects of Heat Shock on Seed Coat Morphology

This experiment aimed to investigate the surface of seed coats of *C. ladanifer* and to assess visually the effects of heat shock on the outermost seed coat integument using as term of comparison the removal of the waxy external layer of seed coats by an organic solvent. Three batches of 10 seeds each were used. Seeds of batch 1 received no treatment. Seeds of batch 2 were stirred 10 min with 5 mL hexane to remove the epicuticular waxes known to constitute the external layer of *C. ladanifer* seed coats [31,41]. After three extractions, seeds were washed with distilled water and dried with blotting paper. Seeds of batch 3 were treated during 15 min with 90 °C in an electric oven (dry heat) and cooled in dry air. Seeds from the three batches were separately mounted on holders with double-sided adhesive carbon ribbon, sputter-coated with gold (JEOL-JFC/1200), observed and photographed using a scanning electron microscope (JEOL, J.S.M. 5200 LV) at 15 kV.

### 4.4. Experiments 4–6: Water Uptake by Seeds

#### 4.4.1. Experiment 4: Weight Increase of Nongerminated Imbibing Seeds

This experiment aimed to investigate whether nongerminated seeds that had not been treated with heat had imbibed. One-hundred seeds were treated with 30 °C as described in Experiment 2, individually weighed to the nearest μg, transferred to four Petri dishes fitted with filter paper, 25 seeds per dish, individually identified, wetted with 5 mL of distilled water and incubated under constant 20 °C with a photoperiod of 8 h. Germination was monitored during five weeks during the light period and germinated seeds removed. Afterwards nongerminated seeds were dried with blotting paper, checked under a hands lens and again individually weighed to the nearest μg. During the second weighing one seed was lost; three were found to be empty from the beginning, 11 lost weight after imbibition (Appendix A). Altogether these 15 seeds were not considered in weight analyses.

#### 4.4.2. Experiment 5: Dye Uptake by Imbibing Seeds

This experiment aimed to investigate which structures of seeds not subjected to heat shock took water during seed imbibition, evaluated by the presence of methyl violet. Four Petri dishes with 25 seeds each were treated with 30 °C as described in Experiment 2 and wetted with 5 mL of 0.5% water solution of methyl violet. An additional Petri dish was wetted with 5 mL of distilled water and used as control. After 72 h excess solution was removed with blotting paper, seeds carefully cleansed with distilled water, dried with blotting paper, sectioned with a scalpel, and the presence of the dye inside seeds visually checked under a stereomicroscope Leica Zoom 2000. Representatives of swollen or non-swollen seeds dyed or not, and of dry seeds used as control were photographed using a Leica DC 300 mounted on a Wild M8 stereomicroscope.

Methyl violet was the second dye used. Following [55] Congo red was used before and never observed inside seeds, presumably because of its tendency to self-aggregation [79,80]. The same executioners following the same procedures sectioned seeds wetted with Congo red or methyl violet. Therefore the absence of the former inside seeds leads us to discard the suspicion that the presence of methyl violet inside seeds could have been a contamination during seeds sectioning with the scalpel.

#### 4.4.3. Experiment 6: Effects of Heat Shock on Volume Increase of Imbibing Seeds

This experiment aimed to compare the effects of 30 °C and 90 °C heat treatments on the percentage of *C. ladanifer* seeds that visibly had their volume increased during imbibition before germination started. Seeds were treated with 30 °C and 90 °C and subsequently incubated as described in Experiment 2. Four Petri dishes per treatment with 25 seeds each were wetted with 5 mL of distilled water. Seeds were examined during 6 days at various intervals under a stereomicroscope Leica Zoom 2000, swollen seeds recorded and discarded. The experiment was continued until no new swollen seed was recorded for two consecutive days.

### 4.5. Depth of Emergence and Soil Temperatures

Depth of maximum emergence of *C. ladanifer* seedlings was determined combining data of hypocotyl growth [26] and maximum depth of seedling emergence estimated using the allometric relationship between depth of successful seedling emergence in soil (*SED*) and seed mass (*W*_s_) expressed in mm and mg respectively [81]:*SED* = 27.3 *W*_s_^0.334^,(1)

Data of published mean values of *C. ladanifer* seed mass [15,24,26,32,82] and of seeds weighed in Experiment 4 were used for that purpose.

Data for soil temperatures during fires were obtained from a variety of sources including published time-course temperature curves [33,34,35,36,37,38] and from simulations using a first order fire effects model [39].

### 4.6. Data Analyses

Comparisons between two means were done by exact or approximate two-tailed Student’s *t* tests with comparison-wise type I error rates of 0.05 after checking for homoscedasticity using the two-tailed *F* distribution with a comparison-wise type I error rate of 0.05. Whenever comparisons among several means were intended, homoscedasticity was tested and when heteroscedasticity was found, Box–Cox transformations [83] were used. In multiple comparisons of means lack of “transitivity” is frequent [84]. To prevent its occurrence a least squares linear regression approach with dummy variables was adopted, and forward stepwise selection with replication was used with candidate models including only the treatments binary coded as (1,0), with an experiment-wise type I error rate of 0.05 for coefficients calculated using Dunn–Šidak method [85]. Lack of fit was always tested and coefficients of determination (*R*^2^) are presented as proportion of the maximum *R*^2^ possible [86]. Likewise when least-squares linear regression was done with measured explanatory variables alone as in the power law fitted to lag of germination in Experiment 2:*l* = *a H*^–*b*^,(2)where *l* is the lag of germination determined from Equation (3), *H* is the heat treatment and *a* and *b* are constants.

Lag, rate, and shape of germination were separately obtained for each replicate in Experiment 2 from the three-term Weibull function [87] fitted by least squares nonlinear regression without replication using the Marquardt method [88]. The three-term Weibull function can be expressed as:*G*_T_ = 1 – exp{–[(*T* – *l*)/*k*]*^c^*},(3)where *G*_T_ is the cumulative germination at time *T* in proportion of final germination registered at the end of the bioassay; *l* (lag of germination) is a location parameter that estimates the latest time at which germination is strictly zero, which in practice represents the time necessary for the first seed to complete germination; *k* (rate of germination) is a scale parameter estimating the rate of germination completion over time with *l*+*k* estimating the time necessary for the completion of 63% of cumulative germination; *c* is a dimensionless shape parameter estimating the symmetry of the distribution of germination over time, with 3.25 ≤ *c* ≤ 3.61 showing symmetry and representing a good approximation to the normal distribution, *c* < 3.25 positive asymmetry, *c* > 3.61 negative asymmetry [89,90]. Fitted equations were only accepted after a consistency check of parameter estimates and germination predictions against the original data [78,91]. Duration of germination (*D*_100_), the number of days necessary to attain final germination minus the number of days necessary for germination to start (*l*) was deduced from fitted equations.

A number of models, including a variety of polynomials were fitted to original data, to means, or to *Q*-splined means [92] and failed to satisfactorily accommodate the whole range of responses of final germination to heat treatments. Therefore regressing final germination on heat treatments was separately investigated for the means of segments 10–70 °C, 70–90 °C, and 90–140 °C by nonlinear regression using a reparameterized Weibull function (Equation (3)) setting *l* = 90. Thus:*G*_F_ = exp – {[(*H* – 90)/*x*]*^z^*},(4)where *G*_F_ is the percentage of final germination at heat treatment *H*, *x* and *z* are equivalent to *k* and *c* in Equation (3).

Data are presented as mean ± SE. Linear regressions, nonlinear regressions and ANOVAs were performed using Statgraphics Plus v. 3.3 (Manugistics, Rockville, MD, USA); determination of Box–Cox transformations using BIOM (Applied Biostatistics, New York, NY, USA); all other statistical analyses using Microsoft Excel^®^ 2010.

## Figures and Tables

**Figure 1 plants-08-00063-f001:**
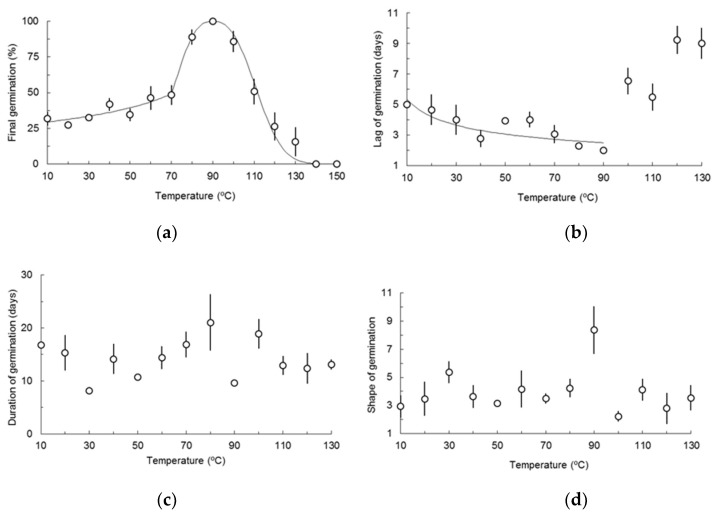
Response of seed germination of *Cistus ladanifer* to heat treatments lasting 15 min: (**a**) final germination; (**b**) time needed for the start of germination (lag); (**c**) time needed to the completion of germination minus the lag of germination (duration); (**d**) shape of germination. Circles for observed values, line for expected values; vertical bars for standard errors (SE).

**Figure 2 plants-08-00063-f002:**
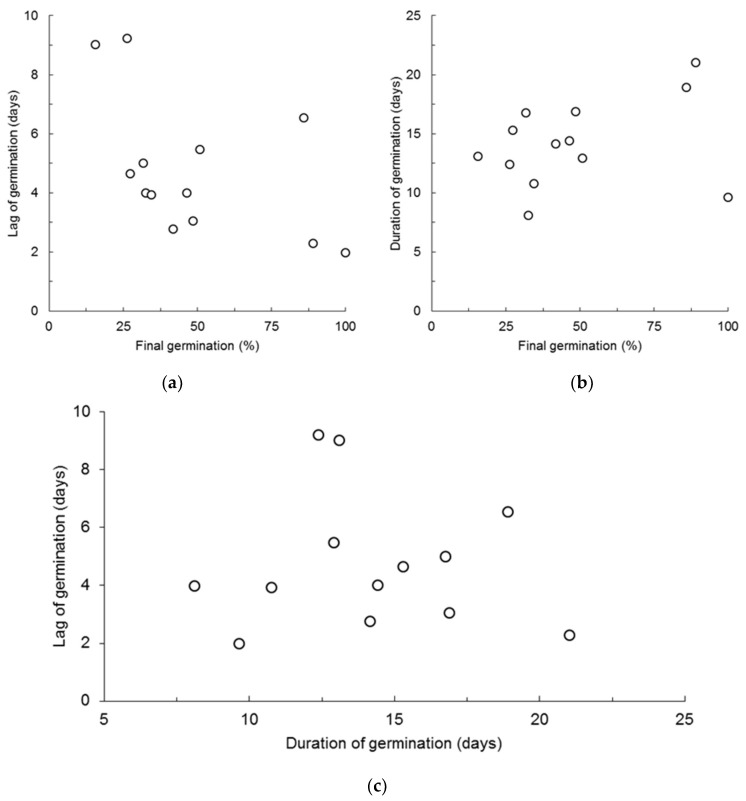
Biplots of: (**a**) final germination and time needed for the start of germination (lag); (**b**) final germination and time needed to the completion of germination minus the lag of germination (duration); (**c**) lag and duration of germination.

**Figure 3 plants-08-00063-f003:**
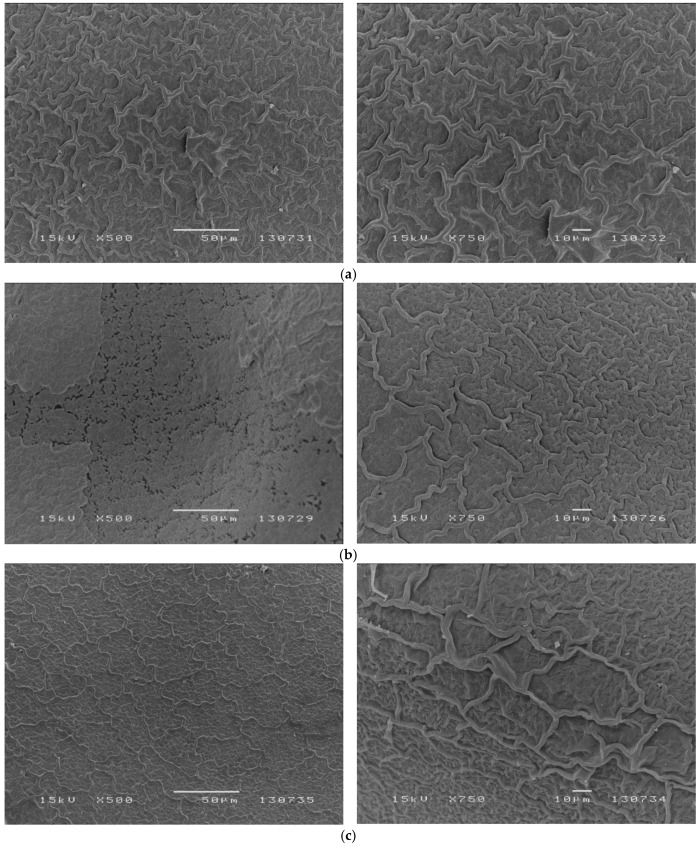
Seed coat morphology of *Cistus ladanifer*. Scanning electron photomicrographs at two magnifications of (**a**) untreated seeds; (**b**) seeds with epicuticular waxes removed with hexane; (**c**) seeds treated with 90 °C.

**Figure 4 plants-08-00063-f004:**
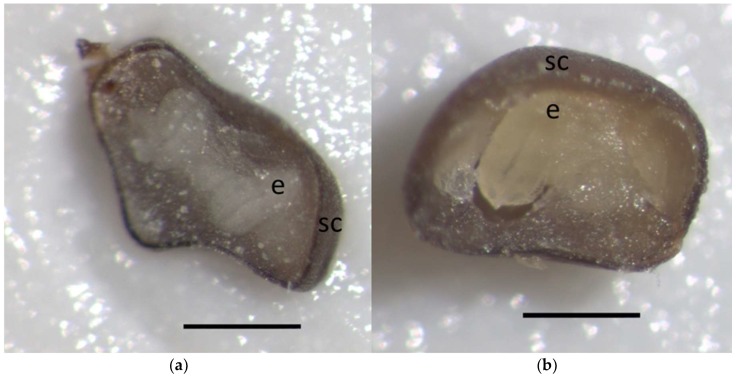
Imbibition and uptake of water and dye by seeds of *Cistus ladanifer*: photomicrographs (**a**) of dry, nonimbibed seed; (**b**) swollen seed imbibing with water alone for 48 h; (**c**) nonswollen seed imbibed with water solution of methyl violet for 72 h; (**d**) swollen seed imbibed with water solution of methyl violet for 48 h. *e*, embryo; *sc*, seed coat. Scale bar = 500 μm.

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
