# Peer review of "Seed Germination in Cistus ladanifer: Heat Shock, Physical Dormancy, Soil Temperatures and Significance to Natural Regeneration"

_plants, 2019, doi:10.3390/plants8030063_

Round 1
Reviewer 1 Report
The authors investigated the effects of different treatments of C. ladanifer seeds on seed germination. The manuscript is well written and good to follow. The introduction nicely leads to the aims of the experiments that were carried out.
In general the manuscript provides new and interesting data on seed germination and dormancy meachanisms.
But I have some main concerns, mainly regarding the structure of the manuscript, that need to be improved and make it in the present form difficult to follow: the individual chapters are not clearly separated, parts in the results belong to the methods, parts in the discussion belong to the results etc. And they are partly redundant. I highly recommend to restructure the manuscript, some major issues are mentioned below. In the present way the manuscript is difficult to follow apart from the introduction.
Results:
70-74: should be in the methods section. Further, the germination percentage for 20 degrees is not shown/mentioned. Maybe a table at this point would be nice to present the data.
81-95: should be removed to big parts into the methods section, except for the results.
In general during the following presentation of the results of the germination trials (figure 1) there is a lot of statistical methodology repeated several times. I assume that it could be summarized as a general procedure and put into the methods section if it is not already completely presented there. In that way it makes the text presentation of the results shown in figure 1 difficult to follow.
141: I do not understand in detail the meaning of `symmetry of the distribution of germination over time´ and what the purpose for measuring it is. Please describe.
2.3 hexane treatment
Actually the results of the hexane treatment with the modified surface only make sense when the effect on germination behaviour followed by the treatment of the seed coat with hexane would be presented as well. Like that it only shows that hexane removes the wax layer, which is not new. To me this part can be removed or a link to the germination has to be presented here.
Lines 176-177: how do seeds in an experiment get lost in a petri dish? I would simply change the number of seeds in the methods section without explaining the reasons (getting lost/being empty).
181-183: this should be in the discussion section.
2.7.1. Depth of emergence of Cistus ladanifer seedlings: I don´t se.e the use in this data. Further it is mainly literature review. To me it does not make sense to present the data at all.
244: Why not writing …to a depth of 50mm, ranging from….? That´s what is shown in figure 4.
249 to 254: This is part of the introduction or discussion, definitely not a result.
322-377: Mainly describes results. I recommend removing big parts to the results section.
359-362: I don´t understand that. You have shown that all seed coats of non- treated seeds are permeable for water without requiring a heat shock, so they are obviously water permeable without any kind of treatment required. Or do I misunderstand this paragraph? Now I see that in lines 419-424 you write that, therefore in my opinion all the discussion before is to the biggest part not needed.
392-399: is part of the methods or results section, not discussion.
In total, the discussion is very lengthy and partly leads away from the actual discussion of the results. I recommend to shorten it to make it easier for the reader to follow.
I miss a final conclusion in the discussion section.
Author Response
Please see upload pdf file.

Reviewer 2 Report
There are two documents attached, the PDF with specific comments on text and a word document with general comments also written below. Please consider both
Interesting piece of research with novel and relevant work for seed ecology area. Mostly clearly written However, there are few inconsistencies and repetitions so organisation is necessary.
General comments
Some statements have several references probably more than the necessary (lines 34-35, 40, 42, 43, 46, 57…) and some have none (lines: 29-30, 32-33…)
Statistic analyses seem to follow the same procedures in several cases (ex. Heteroscedasticity, data transformation), so it could be summarised further not to repeat it every time. Also it should be rather described in the methodology section than in the results and include only relevant and clear information.
The authors should consider that the reader may be reading the paper in the order here and so far nothing indicates that ‘seeds germinating in the 30 °C or 90 °C treatment’ or other temperatures refers to a heat shock treatment that lasts 15 minutes. Rephrasing is necessary that to make that clear in the sections that appear before the methodology
The relevance of experiment 4 is questionable. There seems to be some misunderstanding of seed physiology here. Imbibition is a prerequisite of all seeds for germination so that seeds that remain unimbibed will simply not germinate. Also, imbibition and weight loss at the same time is contradictory per se. If seeds registered lost weight after imbibition then this did not occur and the weight loss may be due to another process as hinted but irrelevant to the objectives of the experiment. Notice that not dormant seeds are expected to germinate in less than a month, as it occurred in most of your seeds with the right heat shock treatment. Furthermore, figure 3 shows according to your reference a well developed embryo that in general germinates well after a treatment that apparently relieves physical dormancy. It is not clear what other type of dormancy could be present. It is likely that seeds that did not germinated could be unviable (probably collected at different maturity stages) rather than present additional dormancy.
Generally an alternative effect of the heat shock has been overlooked. After heat treatment seeds with physical dormancy can become impermeable not only by the rupture of the seed coat as mentioned in this work buy also via the disruption of the lens. Seeds could have relieved dormancy via disruption of the lens with no obvious disruption of the seed coat, as seen in the photos and reported in the results and discussion.

Author Response
Please see upload pdf file.

Round 2
Reviewer 1 Report
The authors addressed all my concerns about the manuscript, the overall structure of the presentation clearly improved and other issues were improved.
I have no further comments.
Author Response
As uploaded PDF file.

Reviewer 2 Report
It is clear that there is plenty of work behind this document. The strengths are the wider ranges of heat shock treatments than other research and the variety of experiments. Most sections are good, yet substantial careful revision needs to be done to be suitable for publication.
I understand the data is old and probably the research has passed from hands to hands until the present and records were lost over the years. Seed collection and experiments started 30 years ago and conducted for at least 20 years. However, there is no evident objective for doing that reflected on the document, that may be as well be conducted on a single year collection considering the results. It seems that a big part of the story is missing, hopefully, to be published too. Furthermore, the methodology does not mention how much of these 20-year data were organized, obtained and grouped to be analyzed statistically. Seed source information missing could be added if available to broaden perspective (ecological, biogeographical or at least as a record among others). Viability, for example, can vary within species, populations, and even the same plant in different years.
There are inconsistencies with the number of replicates and I still cannot see controls. The literature in the previous version was outdated, it has been improved but superficial bibliographic research shows plenty of references about Cistus ladanifer and heat treatments only, conducted in the last decades and many more in other Cistus species. Some of those have been conducted fairly similar experiments and gotten similar results, (in this species, time for germination to start and end after heat shock treatment at different temperatures has already been recorded otherwise). However, the document fails to stress the value of having tested a wider range of temperatures or use the existing references to support the research.
Methodology and the objectives tend to be rather subjective. It was partly improved from the last version, but it is not clear how most things were done and why. Part of those is missing perhaps because they are obvious to the authors but are not for the reader and not described to be replicable.
There is no clear path from methods to results exposed or hint of replicability. There is no consistency on what is written in the methodology (replicates, treatments and species pooled or treated otherwise), and results. In the previous revision, it was noted as a general comment but looks not to be understood.
The responses for experiment 4 are well defended in the comments the to reviewer in the sense that it lets clear that seeds can lose material during imbibition. It shows familiarity and understanding of relevant literature. However, it is not so in the document, the arguments are not as strongly defended there. The rationale of weighting ungerminated seed after all other had germinated and not conducting any type of viability test at any stage at least to discard it as factor for lack of germination is not clear. Viability was inferred and extended from other results but not tested at any stage. Imbibition occurs commonly in a portion of most seeds of species described as PY as in your results but is not exclusive of viable seeds. Unfortunately, without a test, results can only be inferred and can be biased towards the desired results.
By the way, the water gap referred before opens by itself with heat, is not separated process, it just happens when heat is applied. The comment before was so that the process was not overlooked, and the authors could consider complementing the discussion. Comments are not meant to change the direction of an experiment that was long ago finished.
Other suggestions:
Avoid imprecise expressions such as much more, clearly, undoubtedly, evidently etc. Scientific literature requires objectivity.
The number of references 99 seems excessive for the paper, several are hardly relevant.
The book of Baskin and Baskin 2014 has tables and databases with references to most seed publications in any area until recently. Cistus species, dormancy and treatments to break it are well referenced. Needless to say, it is a valuable resource by itself.

Author Response
As uploaded PDF file.

Round 3
Reviewer 2 Report
The current arrangement of the document reads clear the relevance of the research. Most comments have been addressed and after considering a few edits below and those the editor may consider, I advise this research meets requirements for publication.
Attached is the cover letter with replies to the author and below an issue that I consider relevant to sort at least in writing. The remaining are mere observations and minor changes proposed.
Main issue:
The frame shows disagreement between your research that suggests no PY present in that species and others’ vague observations presuming that there is PY. I write vague based on your statements. It is noticeable that you know your population well and for a long time, but it seems much to generalise to all the species the lack or presence of PY dormancy with your current data of a single population.
It can be argued that in dormant species, dormancy is present in different degree among populations, gradients, or even within populations. This does not necessarily mean that dormancy is (or not) a characteristic of the species, but it might be related to a particular population and its environment and in this case to collection time, storage conditions, seed age when tested, population genetics etc. All considering this is a widespread species and your population must be in an edge of its natural distribution. However, your contributions recording your population presumed lack of PY dormancy is relevant enough.
If heat is applied at the right intensity to PY dormant seeds, heat can either break the seed coat or open the water gap and the result will be in both cases the relieve of PY dormancy. In the experiment, you considered only the damage to the seed coat and overlooked the possibility of the other effect. How are you so sure that the water gap did not open with heat and relieved PY dormancy in those few presumed viable seeds that would not germinate without heat treatment? It would be expected both potential effects of heat on PY were considered to discard it conclusively.
In this research, without treatment not all seeds imbibe, but after heat treatment seeds germinate up to 100 % at 90°C. At a first glance that suggests PY as in other research. Yet, there is still a small difference between almost all and all, but that relates more to the presumed 100% seed viability assumed, and not to the methods that in any other unluckier case (with less presumed viability) could be questionable.
Others
Viability test should be used to dismiss it from other factors affecting your results. You might have experienced that 100% viability is not the most common situation from seed collection in the wild. It would have added a plus to the individual seed analysis that you conducted.
Discussion on the paper lacks the strength that authors use in discussion to reviewer’s comments. These last are better defended and referenced than those on the document.
Assuming all seeds are permeable and imbibe with no treatment, and your broad experience on the subject I would expect this paper lay the foundations giving some insights to the direction to follow in future research.
Lines 91 and 145: 15 minutes
Line 554: add references
Line 558: review references
Line 686- 687: this is an example that you have got no conclusive evidence as in other research you consider invalid for reference
Line 688: to be only one of the at least two pre-requisites for water uptake
Lines 694-695: increased germination using solutions is not related to this work
Line 699: I guess this should say that dormancy if present should not be PY
Lines 692, 701, 705, 708, 721: there are data figures (% for example) to write here complementing or instead of almost all, with few exceptions or around.
Line 872: the 29 years 1971-2000 goes with collection data lines 869-870
Lines 869 and 882: delete always
Line 955: delete above
Lines 965-967: Such…C. ladanifer in the discussion.

Author Response
Please see the pdf attachment.
